# Consequences of the COVID-19 Pandemic and Governmental Containment Policies on the Detection and Therapy of Oral Malignant Lesions—A Retrospective, Multicenter Cohort Study from Germany

**DOI:** 10.3390/cancers13122892

**Published:** 2021-06-09

**Authors:** Diana Heimes, Lena Katharina Müller, Alexandra Schellin, Hendrik Naujokat, Christian Graetz, Falk Schwendicke, Maximilian Goedecke, Benedicta Beck-Broichsitter, Peer W. Kämmerer

**Affiliations:** 1Department of Oral- and Maxillofacial Surgery, University Medical Center Mainz, Augustusplatz 2, 55131 Mainz, Germany; lena_katharina.mueller@unimedizin-mainz.de (L.K.M.); peer.kaemmerer@unimedizin-mainz.de (P.W.K.); 2Department of Oral and Maxillofacial Surgery, University Hospital of Schleswig-Holstein, 24105 Kiel, Germany; a.schellin@web.de (A.S.); hendrik.naujokat@uksh.de (H.N.); 3Clinic for Conservative Dentistry and Periodontology, School of Dental Medicine, Christian-Albrechts-University of Kiel, 24118 Kiel, Germany; graetz@konspar.uni-kiel.de; 4Department of Operative and Preventive Dentistry, Charité University of Berlin, 10117 Berlin, Germany; falk.schwendicke@charite.de; 5Department of Oral- and Maxillofacial Surgery, Charité-Universitätsmedizin Berlin, 10117 Berlin, Germany; maximilian.goedecke@charite.de (M.G.); benedicta.beck-broichsitter@charite.de (B.B.-B.)

**Keywords:** COVID-19, SARS-CoV-2, oral cancer, oral malignant lesions, screening, precursor lesions, lockdown, pandemic, head and neck cancer, HNSCC

## Abstract

**Simple Summary:**

Due to the COVID-19 pandemic, the oncology community was challenged with the need to protect a vulnerable population from a potentially fatal infection without jeopardizing cancer treatments. The impact of the crisis on the medical care of patients with oral cancer is largely unexplored. This multicenter cohort study from Germany aims to assess the consequences of the COVID-19 pandemic by comparing the healthcare of patients during the lockdown and post-lockdown periods in 2020 with the corresponding periods in 2018/19. We found the closure of dental practices during lockdown to possibly delay the diagnosis of oral cancer. Even if during this period no higher incidence of oral cancer was observed, data point to potentially fatal consequences for longer periods of treatment delay.

**Abstract:**

(1) Background: In response to the global COVID-19 pandemic, governmental measures have been undertaken. The impact of the crisis on the healthcare of patients with cancer is largely unexplored. This multicenter cohort study aimed to investigate a potential screening delay and its consequences in patients with oral cancer (OC) during the pandemic. (2) Material and Methods: Data of patients who were first diagnosed with OC during different periods were collected, especially in terms of OC incidence, tumor stage/entity and time to intervention. The periods lockdown (LD) (13 March–16 June 2020), post-lockdown (PLD) (17 June–1 November 2020), and the corresponding equivalents in 2018/19 were differentiated and compared. (3) Results: There was no obvious trend towards a higher incidence of OC or higher tumor stages, whereas a trend towards a shorter time to intervention during the LD2020 could be observed. Subgroup analyses revealed an increased incidence in OC within the PLD2020 in Mainz, which might be explained by the partial closure of dental practices in this federal state during LD. (4) Conclusions: While there was no overall higher incidence of OC, we found closure of practices during LD to possibly delay cancer diagnosis. Therefore, measures must be taken to identify patients at risk and to ensure basic healthcare, especially in the context of dental screening measures.

## 1. Introduction

The COVID-19 pandemic, with over 100 million people infected and 2 million deaths so far [1], is often compared to other pandemics in world history. Between 1918 and 1920, the Spanish flu—the worst influenza pandemic in history—killed between 27 and 50 million people. However, what is the difference between these two pandemics? Coronaviruses are ribonucleic acid (RNA) viruses of which some can infect humans causing a wide range of diseases including gastrointestinal, respiratory, and central nervous system disorders. SARS-CoV-2 was first described in China in late 2019 and is zoonotic in origin—bats have been suggested to transmit the virus to humans [2]. Coronavirus disease is characterized by an influenza-like illness including fever, cough, and myalgia; olfactory and taste disorders were described frequently being a rare symptom in other respiratory virus diseases [3]. In addition to the type of virus, external circumstances such as malnutrition, hygiene, access to medical care, and the scientific standards of the time caused a much higher mortality rate. Nevertheless, even about 100 years ago, there were calls for public health authorities to work together with the medical profession and for a “public health authority organization” to prevent the disease from spreading further [4]. Today, in the 21st century, such circumstances are hardly conceivable in industrialized countries. In addition to excellent national medical care, state regulatory mechanisms ensure containment in pandemic situations. To cope with the current COVID-19 pandemic, strategies of different escalation levels are applied with the aim of containment, protection and risk reduction. A nationwide “lockdown” was therefore declared on 16 March 2020, in Germany to prevent the health system from being overloaded. Hospitals and practicing dentists equally had to adapt to the state of the pandemic. Elective hospital admissions were stopped, non-emergency operations were postponed and in some federal states, dentists were allowed to treat emergencies only. This disruption of the regular care structure put oncological patients with oral malignant lesions at increased risk, as screening examinations were suspended, and urgent symptomatic cases were prioritized for diagnostic intervention. Kranz et al. conducted a survey within the US and found a delay in receiving dental care in nearly half of the population (46.7%) due to the COVID-19 pandemic [5]. Since oral precursor lesions and oral malignant lesions are mainly detected by dentists during routine screening, it is reasonable to assume that the nationwide lockdown and its restrictions on healthcare could have led to failures in the screening of high-risk patients and consequently to an increase in avoidable malignant lesions as well as a delayed diagnosis of oral tumors. Arduino et al. found such a delay in diagnosis of oral cancer in the north Italian area of Turin due to the SARS-CoV-2 outbreak [6]. Accordingly, not only an increased number of oral malignant lesions after the lockdown but also the occurrence of higher tumor stages in the future would have to be expected.

Oral cancer is among the 10 most common cancers in the world [7] and early diagnosis significantly decreases morbidity and mortality [8]. About 5% of all malignant neoplasms affect the oral cavity. Histologically, squamous cell carcinomas are present in 87% of cases, and adenocarcinomas in about 4%. In Germany, men are affected more frequently and on average three years earlier than women. Here, the incidence among women is about 4500 per year, among men more than 9000. According to the UICC tumor stage data, more than one in four to five tumors in women is diagnosed at an early stage (0/I), while the same number for men is every seventh. This also causes a significantly higher mortality rate in men (9.7/100,000) when compared to women (3.5/100,000). The most important risk factors for development of an oral malignant lesion are all forms of tobacco and alcohol consumption. If both factors act together, the effect is considerably amplified [9]. Another major risk factor is chronic infections with human papillomavirus (HPV) and the presence of oral precursor lesions [10]. Since the prognosis for early-stage tumors is much better than for advanced stages, early detection is of central importance and several early detection methods and techniques have been advocated [11,12,13,14]. Even so, at present, there is no specific biomarker to detect precursor lesions and early stages of oral cavity carcinoma. By definition, a biomarker shows that a particular biological process or condition is present. In cancer diagnostics this means any measurable indicator used to determine the existence of the disease. Since oral squamous cell carcinomas (OSCC) typically originates from non-aberrant keratinocytes which are chronically exposed to a toxic stimulus leading to dysplasia followed by invasive growth of carcinomatous cells, carcinogenesis consists of a smooth transition that is sometimes hard to detect or to differentiate from benign lesions [7]. Latest studies could show that increased levels of interleukin-8, subcutaneous adipose tissue and pSTAT3 can be found in patients with OSCC [7,15]. Baran et al. found an association between the melanoma-associated antigens A family and a high risk for malignant transformation of leukoplakia [16]. Furthermore, recent evidence shows some chromosomal changes that could be associated with OSCC. However, those findings are still far beyond a specific biomarker that can be used in early detection of oral malignant lesions. Most diagnostic measures still rely on surgical sampling which is an invasive method that cannot be used as a general screening parameter for the population and therefore requires a qualified specialist to identify and closely monitor and, if necessary, initiate further diagnostics for patients at risk [17]. Since the growth of tumors is often painless in the beginning without functional impairments, patients typically consult a specialist with a delay of several weeks. The misinterpretation of lesions or their relevance are supposed to be the main reasons for treatment delay in oral cancer. For this reason, regular screening by the dentist is of crucial importance, especially for patients with risk factors or predisposing diseases of the oral mucosa [9].

During the early period of the COVID-19 pandemic, especially between March and June 2020, a reduced number of cancer diagnoses were reported by countries around the globe [18,19,20]. Lai et al. stated a 66% reduction in chemotherapy administration across eight hospitals in the UK [21], in Germany, radiation oncology institutions were able to perform curative procedures with either unchanged or moderately decreased schedules, whereas palliative or other radiotherapy procedures were often postponed [22].

A national population-based modeling study from the UK estimated a substantial increase in the number of avoidable cancer deaths in the UK considering four tumor types (breast cancer, colorectal cancer, lung cancer, and esophageal cancer). Based on 113,607 tumor cases, the group considered different reallocation scenarios that allowed the assumption of more than 3000 additional deaths due to a diagnosis delay of 12 months over a 5-year period [23]. Jacob et al. conducted a retrospective study on over 100,000 patients in Germany newly diagnosed with cancer between May 2019 and May 2020. They found a significant decrease of cancer diagnoses in general practices between March and May 2020 of up to 27.6%, and a similar trend in specialized practices with fewer initial diagnoses of 28.2–44.4%. This was pronounced among cancers of the skin and intrathoracic organs [24]. Another study analyzed cancer reporting by pathologists in North Rhine-Westphalia (Germany) showing a relative reduction of 21% in cancer cases reported compared to the previous year. Interestingly, the authors reported that there was no evidence of a “catch-up effect” following this temporary decrease up until September 2020 [25]. Those findings are in accordance to analyses from other European countries reporting a 44% reduction in newly diagnosed cancer in Belgium during lockdown in April 2020 [26] and a weekly increasing reduction of cancer diagnoses in the Netherlands of 9% up to 26% after the beginning of the nationwide lockdown on March 15, 2020. This effect was especially pronounced in skin cancer diagnoses showing a decrease of 44% up to 60% (16th through 19th calendar weeks) [25,26].

Reasons for this may be a reduced participation in screening programs, a delayed detection of symptoms due to general practitioners’ practices being closed, or a delayed treatment initiation even for patients already diagnosed with cancer due to reduced heath care capacities [18]. A study by Riemann gave evidence that—in Germany—contacts with the primary care physician dropped by up to 70% during the lockdown in 2020. In addition, gastroenterologists experienced a massive cancellation of appointments for elective examinations and a reduction in the number of appointments for preventive examinations [27].

Up to now, there is no therapeutic approach available to treat COVID-19. Interestingly, nanomedicine-based strategies—also used in targeted cancer therapy—have been proposed to be a promising option to both, treat viral infections and regulate the host’s immune response [2]. Though, since the COVID-19 pandemic is still in progress and is likely to reoccur in the future, there is a great need for evidence-based guidelines to ensure healthcare, especially for patients with chronic diseases in compliance with infection preventive measures.

Up to date, there has been no scientific evaluation on the impact of the COVID-19 pandemic and its restrictions on healthcare on oral cancer detection. Therefore, this study aimed to investigate the impact of the COVID-19 pandemic on patients with oral malignant lesions

## 2. Materials and Methods

### 2.1. The COVID-19 Pandemic in Germany

Germany reported the first confirmed SARS-CoV-2 infections in January 2020, and by mid-February 2020, multiple infection clusters had formed. To minimize the expansion of clusters, the German disease and epidemic control advised by the Robert Koch Institute (RKI) managed the outbreaks in the so-called protection stage from 13th of March 2020: academic semesters were postponed, kindergartens were closed and dentists were, in some parts of Germany, advised to either close their practice or to perform emergency treatments only. Consequently, the Government declared a state of emergency, which was revoked on 16 June 2020. On November 1st, restriction measurements were reimplemented due to rising infection rates. The analytic periods in this study were chosen along this course of the pandemic in Germany. After the incidence-triggered lockdown ended, patients with postponed procedures were triaged and scheduled for therapy, including elective procedures, according to their individual risk of infection and the urgency of the procedure, in order to avoid the massive accumulation of scheduled procedures in subsequent months and years.

### 2.2. Data Sources

Electronic medical records from the oral and maxillofacial surgery departments in Mainz, Kiel and Berlin were queried using the ICD-10 diagnosis codes and those patients who met the inclusion and exclusion criteria were identified. The three hospitals chosen are located in three different federal states within Germany. In Mainz, located in Rhineland-Palatinate, dental practices had to be partially closed for elective treatments during the first lockdown, whereas in Berlin, federal state of Berlin, and Kiel, federal state of Schleswig-Holstein, there were no closures required.

### 2.3. Inclusion and Exclusion Criteria

Patients with a histopathological confirmed diagnosis of oral cancer (ICD-10 code: C 00–C 14) who were initially diagnosed within the defined periods were included in the study. The interval between 13 March 2020 and 16 June 2020 was defined as “lockdown”; accordingly, the interval between 17 June 2020 and 1 November 2020 was considered the “post-lockdown” period. The corresponding equivalents in 2018 and 2019 were used as control groups.

Exclusion criteria were initial diagnoses outside the defined periods, extraoral cancer, precursor lesions as well as extraoral manifested recurrences such as lymph node metastases and distant metastases. Other factors that led to exclusion from the study were lack of adherence to therapy (therapy delay, bias in time to intervention), death of the patient within the period between the initial diagnosis and the start of therapy, the need for further therapeutic interventions if the patient was at vital risk from other diseases (e.g., bypass surgery for cardiovascular diseases) between the initial diagnosis and the start of therapy, as well as dropout and lack of follow-up.

### 2.4. Outcomes

The primary endpoint of the study was the incidence of patients with an initial diagnosis of oral cancer within the different periods. Furthermore, the cancer stage according to the TNM-classification and the UICC-classification was recorded. Analysis of time to the intervention was also defined as the primary endpoint. Various factors were evaluated for their independent association with an overall oral cancer diagnosis including the specific type of diagnosis according to the ICD-10 classification, type of treatment (surgery, radiotherapy, chemotherapy, immunotherapy, palliative therapy) and the form of diagnosis (histopathological initial diagnosis after internal sampling and evaluation or after collection by an external physician and sending to an external institute for processing; external vs. internal).

### 2.5. Statistics

The statistical analyses were performed using SPSS version 27 for Macintosh (IBM, Armonk, NY, USA). To analyze the differences between the measured values, normality (Shapiro–Wilk) and homogeneity of variance tests (Levene Statistic) were performed at first to check the conditions for the subsequent analysis. In case of not normally distributed values, a Mann–Whitney test was used. To investigate whether the different tests lead to similar results, McNemar tests were performed. A *p*-value *≤* 0.05 was termed significant.

## 3. Results

A total of 653 cases from three departments that met the inclusion criteria for this study were analyzed. The different tumor entities according to ICD-10 coding, as well as the tumor stages (TNM, UICC), kind of treatment modality, and kind of diagnosis are shown in Appendix A.

### 3.1. Multicenter Data Analysis

The pooled analysis resulted in a total of 653 cases that met the inclusion criteria. A total of 228 cases were identified for 2018, of which 77 were in the lockdown equivalent (LD-E) period and 151 in the post-lockdown equivalent (PLD-E) period. The numbers in the following years remained stable with a total of 210 cases in 2019 and 215 cases in 2020. The proportion between the different periods was also relatively stable at about 34–40% during the LD(E) period and 60–66% during the PLD(E) period. Regarding the UICC stage ratio, a relatively stable distribution was seen with a proportion of 50% high tumor stages in the LD(E) period while slightly fewer higher tumor stages were diagnosed in the PLD period in 2020 compared to previous years (50.79% versus 59.18%). The analysis showed a trend towards a shortened time to intervention during the 2020 LD period when compared to previous years (22.99 ± 12.72 d versus 26.66 ± 14.29 d) while the time in the 2020 PLD period remained stable compared to previous years (26.05 ± 13.84 d versus 26.14 ± 15.14 d). The multicenter data analysis confirmed the impression of a shorter time to intervention by internal diagnosis, especially during the LD period in 2020 and the equivalent period in 2018 and 2019 (LD 2020: *p* = 0.026, 21.05 ± 12.29 d vs. 28.79 ± 12.54 d; LD 2018/2019: *p* = 0.026; 25.10 ± 15.21 d vs. 30.15 ± 11.37). Besides, a relatively shorter time to intervention during LD 2020 could be observed, whereas values did not differ much within the other periods (Figure 1).

### 3.2. Subgroup Analysis: Medical Department Mainz

The data analysis yielded a total of 160 cases that met the inclusion criteria. In 2018, 54 patients were found, 14 of whom could be assigned to the LD-E period and 40 patients to the PLD-E period. In 2019 there were 25 patients in each period. In 2020, 20 patients could be assigned to the LD period and 36 to the PLD period. The LD- to PLD-ratio was 25.93% to 74.07% in 2018, 50% in each in 2019, and 35.71% to 64.29% in 2020. As such, there is no obvious trend in 2020 compared to previous years in terms of patient number distribution.

Concerning the UICC classification, a trend towards a higher percentage of smaller cancer stages was evident when comparing the LD period in 2020 to the reference years 2018 and 2019 (UICC I: 45% versus 28.21%). In 2020, just minor changes in UICC stadium distributions were seen when compared to the PLD-E period in the reference years. The subgroup analysis also found more patients with lower tumor stages (UICC I and II) within the LD period in 2020 compared to the previous years. Interestingly, the time to intervention (regardless of the type of intervention) in the 2020 LD period with an average of 19.24 ± 993 days was lower than in the equivalent period of previous years (22.97 ± 10.29 days). With 25.44 ± 11.39 days, the time to intervention in the PLD period could be shown to be close to that of the previous years (26.90 ± 19.28 days). There was a statistically significant difference in the time to intervention between the LD and PLD period in 2020 (*p* = 0.04). Interestingly the time to intervention differed remarkably between internal diagnosis and external diagnosis with a statistically significant shorter time interval for internal diagnosis (14.17 ± 4.84 d vs. 26.88 ± 10.95 d) during LD 2020 (*p* = 0.013). Contrary, time to intervention for internal diagnosis increased again from LD to PLD (*p* < 0.001, 14.17 ± 4.84 d vs. 25.13 ± 11.51 d). It is this large reduction in time to intervention for internal diagnosis in LD 2020 that probably causes the significantly lower time to intervention overall compared with the other periods, which remained relatively stable in comparison with each other (Figure 2).

### 3.3. Subgroup Analysis: Medical Department Kiel

The analysis yielded a total of 167 cases that met the inclusion criteria. In 2018, there were 64 cases in total, 30 in LD-E, 34 in the PLD-E period. In 2019, fewer patients were initially diagnosed with oral carcinoma in the indicated periods (n = 47), and in 2020, 56 were diagnosed, including 21 in the LD period and 35 in the PLD. Here, the percentage of patients in the 2020 LD period was lower than that within the other periods (2020: 37.5%, 2019: 42.55%, 2018: 46.88%). Compared with previous years, 2020 showed a significantly higher proportion of stage IV carcinomas (50% versus 31.71%) in the LD period, while the proportion of stage I carcinomas decreased significantly (18.75% versus 36.59%). In contrast, the number of high tumor stages in the 2020 PLD period was similar to previous years (51.72% versus 45.1%). Overall, there was no significant change in time to intervention in the 2020 LD period compared with previous years (29.85 ± 11.67 d versus 30.49 ± 12.53 d) while it was slightly, but not significantly, lower in the 2020 PLD period compared with 2018 and 2019 (26.65 ± 13.97 d versus 30.63 ± 11.25 d). Furthermore, no statistically significant difference could be shown in time to intervention between internal and external diagnosis. The time interval remained relatively stable between 25 and 35 days (Figure 3).

### 3.4. Subgroup Analysis: Medical Department Berlin

The analysis for the medical department Berlin resulted in a total of 326 cases that met the inclusion criteria. For 2018, a total of 110 cases were identified, of which 33 fell into the LD-E period and 77 into the PLD-E period. A similar distribution was seen in subsequent years, with a total of 113 first diagnosed oral cavity carcinomas in 2019 and 103 in 2020. The distribution across the respective periods was relatively homogeneous at approximately 30–34% within the LD-E period and 66–70% during the PLD-E period with a slightly higher proportion of patients in LD 2020 of 36.89%. Regarding the distribution of UICC stages, there was a slightly higher proportion of high tumor stages in the LD period 2020 compared to previous years (52.78% versus 47.89%), with this shift mainly due to a higher number of initial diagnoses of stage III cancer. In contrast, the PLD period showed a significantly lower proportion of diagnoses of higher tumor stages compared to previous years (54.1% versus 72.67%). Within the 2020 LD period, a slightly lower time to intervention was measurable compared to previous years (23.06 ± 12.61 d versus 25.68 ± 16.7 d), whereas the data did not differ in the PLD period (25.28 ± 14.52 d versus 25.00 ± 16.55 d). Within the PLD period in 2020, the time to intervention for internal diagnosis was significantly lower than that of the external diagnosis (*p* = 0.024; 22.38 ± 14.71 d vs. 30.81 ± 12.72 d). Although the analysis was able to show quite homogeneously a shorter time to intervention within the group of internal diagnoses, no other significant differences were found, possibly resulting from a relatively high standard deviation arising from a small group size combined with a great heterogeneity of the group (Figure 4).

## 4. Discussion

The COVID-19 pandemic, now lasting for more than a year, is going to affect public health and the entire society in the long-term. Efforts are made to lower the number of people infected with Sars-CoV-19 while maintaining high-quality standards in the treatment of other diseases. This relies on an efficient healthcare system adopting a double strategy. Furthermore, not only the patient’s safety has to be guaranteed, but also healthcare professionals have to be protected from the infection as they are both of systemic importance and play a major role in limiting the spread of the disease. Given the transmission mechanism via respiratory droplets [28], head and neck surgery has been recognized as one of the specialties at the highest risk since the onset of the COVID-19 pandemic due to the invasive nature of examinations with patients not being able to wear a mask and the aerosol-generating procedures like ultrasonic scaling. In a multicentric survey (23 maxillofacial surgery departments) from Italy, 4% of maxillofacial residents tested positive for SARS-CoV-2 [29]. Reports like this caused an energetic discussion regarding the necessary measures to be taken in dental practices and clinics. During the first pandemic wave, Germany and other countries put a general lockdown in place more including the closure of dental offices in some federal states, such as Rhineland-Palatinate. On the other hand, a disproportional higher risk of being harmed by the coronavirus of elderly patients and those with comorbid conditions is known. Reports from China point to a higher risk for significant morbidity, including requirements for ventilatory support with a hazard ratio of 3.56 for patients with cancer who acquired COVID-19 [30]. What is more, cancer patients on active therapy require more interactions with healthcare providers further increasing the risk of exposure to SARS-CoV-2 besides a higher vulnerability due to oncologic treatments like surgery and especially radiation and chemotherapy [3,31,32]. Another point to take into account is that cancer patients are generally reported to be older and to have more comorbidities than the non-cancer cohort [3]. To address the higher risk of cancer patients for both, exposure to SARS-CoV-2 and a prolonged and severe coronavirus infection, the Infectious Diseases Working Party (AGIHO) of the German Society for Hematology and Medical Oncology (DGHO) prepared a guideline on evidence-based management of COVID-19 in cancer patients. They recommended to not delay/discontinue radiotherapy, targeted therapy, endocrine therapy or surgery in patients without SARS-CoV-2 infection, whereas a delay/discontinuation of chemotherapy and surgery were strongly recommended in patients with COVID-19 as chemotherapy was reported as a risk factor for severe COVID-19 and a perioperative infection was associated with a higher mortality rate in patients treated by surgery [3]. The American Cancer Society estimated an amount of 5000 new cancer diagnoses per day in 2021 in the US, which highlights the relevance of a well-structured healthcare system during the pandemic to (1) balance the risk of a COVID-19 infection and the potential delay in cancer diagnosis and treatment, and (2) manage the allocation of limited healthcare resources during the healthcare crisis [31].

Zhang et al. reported a dramatic decrease in hospital visits to 6% compared to the previous years in a hospital in Hangzhou (China). However, emergency visits were increased 16-fold [33]. A multicentric survey performed 4 weeks after the pandemic started in Italy reported of an 87% decrease in outpatient visits and surgery, a reduction in general anesthesia surgery of 78% countrywide, whereas—although with a reduced incidence—the surgical management of trauma and oncology have been carried out [29]. An overall reduction of outpatient visits of 50% per week due to the measures implemented to contain the pandemic was seen in a university medical center in Germany [34]. What is more, a temporary interruption of private practice was observed in order to protect healthcare workers and patients [29]. With dental offices reopening after the lockdown period, strict protocols for patient triage and staff protection were implemented. Given that the pandemic situation is still ongoing, these measures continue to be recommended, with economic and public health consequences of massive extent. Preventive measures aim to slow down the virus transmission via social distancing, hygiene, face masks, testing, isolation, and the avoidance of public crowded places [35]. The following suggestions have been made for the management of dental appointments: only patients with appointments should enter the dental office; ideally, one patient is allowed to sit in the waiting room only. Patients should not be accompanied unless necessary. Contact with a suspected/confirmed COVID-19 person within 14 days or a positive test for COVID-19 should lead to a reschedule of the appointment unless it is a dental emergency. Those patients should be treated minimally invasive without aerosol-generating procedures if possible; airborne precautions have to be added when this cannot be avoided. The university cancer center Hamburg recommended to reschedule routine follow-ups of patients without active chemotherapy until new symptoms are reported. Outpatient care of patients with active treatment should be continued as planned. Here, especially radiotherapy of cancer patients in curative intention should not be interrupted, whereas other radiation schedules should be reviewed for their prioritization [34]. Tele-dentistry/medicine was suggested as an alternative for face-to-face outpatient visits [35,36]. In consideration of these various measures, including the closure/partial closure of dental offices during the lockdown in some federal German states and the limitation to the treatment of dental emergencies, it would not be surprising to observe an increase in the number of oral malignant lesions after the lockdown period compared to previous years.

In our study, the multicenter analysis showed no significant difference regarding initial tumor diagnosis during LD and PLD compared to the previous years. Due to the relatively limited number of patients, a reliable conclusion, especially retrospectively, was naturally difficult, but there seems to be an absolute increase in early tumor stages (I and II) in the PLD period in the medical department Mainz compared to the pooled mean of the previous years. This could be attributed to an increased number of malignant transformations of precursor lesions due to the delay in screening, which could eventually be a result of the closure of dental practices within the LD in Rhineland-Palatinate, which did not occur in the federal states of Kiel and Berlin. There, dental practices were allowed to treat patients without any restrictions during LD. In accordance to this, the other medical departments showed a trend towards a constant number of patients with a redistribution to the PLD in Kiel and even a reduction in the total number of patients in Berlin with a constant distribution to the different tumor stages compared to previous years.

On the one hand, the postponement of regular preventive examinations—especially in patients with precursor lesions, but also in the regular care—could have led to a delay in the diagnosis of malignant lesions; on the other hand, individual cases allow the assumption that patients’ fear of infection also led to a delayed visit to the dentist, and thus higher tumor stages may be observed after the lockdown than would normally be expected. In this study, neither an overall increase in cancer diagnoses nor higher tumor stages could be observed for the lockdown and the post-lockdown period compared to the previous years. Contrary to the initial suggestions, time to treatment decreased during the lockdown period (22.99 ± 12.72 d vs. 26.05 ± 13.84 d), presumably as a result of cancer diagnoses being characterized as “essential surgery” that cannot be delayed. Furthermore, more hospital staff have been available during the LD since elective surgeries were postponed. Institutional protocols require the specification whether a surgical procedure will require postoperative intensive care unit (ICU) admission (planned or inadvertent complications); the decision to undertake such a procedure must be considered in conjunction with availability of ICU beds, therefore, a postponement of essential cancer surgery due to a lack of ICU beds would not have been surprising. Overall, the limit of available ICU beds was not reached, and beds were still available during the first LD in Germany.

The relationship between the time to treatment initiation and health outcomes has extensively been explored. In a systematic review, Neal et al. summarized the effects of a delayed treatment for different cancer entities, including oral cavity carcinoma, and found a significant association between a shorter time to first diagnosis and a better treatment outcome [37]. According to the data published by Goldsburry et al., the economic impact of treatment delay in terms of healthcare cost weighted according to the distribution of disease stage at treatment initiation showed an average increase in treatment costs of about 250% between stage I and stage IV of the diseases (according to the publication exemplified by the tumor entities breast cancer, colorectal cancer, lung cancer and melanoma) [38].

Based on Australian data for four of the country’s most common cancers, Degeling et al. developed an inverse stage-shift model to estimate the excess mortality and health economic impact of delayed access to cancer services due to the COVID-19 pandemic and found a calculative probability of progression from stage I to stage II in lung cancer for a complete delay of 3 months of 8.3%. For a 6-month delay this probability was estimated to be 16%. The model estimated that this would result in 11 and 43 excess deaths after 5 years within the population surveyed. The authors estimated an average increase in healthcare costs, depending on the cancer entity, of up to AUD 36 million (6-months delay) [18]. Sud et al. calculated an average loss of 0.97/2.19 live-years gained per patient for a 3/6-month delay in cancer surgery in England. With 94 912 resections of major cancers per year, this would result in 92 214/208 275 life-years lost due to the treatment delay [39]. Head and neck cancers are reported to be one of the fastest growing tumor entities with an estimated doubling time of 99 days. On average, new metastases were detected after 14.9 months leading to upstaging of the tumor. A 4.8% increase in tumor control risk was estimated for a two-month treatment delay, whereas six months of delay are thought to lead to an increase in risk of 6% [40]. In a large retrospective cohort study with over 37,000 patients included, Rygalski et al. showed that a surgical delay of longer than 67 days independently increased the risk of death (hazard ratio 1.189). The authors suspected a potential link between an increased time to intervention and the risk of distant metastases as well as perineural and lymphovascular invasion and greater depth of invasion [41]. In consideration of those numbers, the dentists’ unique role as point of care facility for patients with oral malignant lesions and as a system-relevant institution for early diagnosis and therapy initiation becomes evident. Assuming that in about 8% of carcinomas there is a progression from stage I to stage II within 3 months, there should be a mathematical excess of 17 patients with a higher stage carcinoma in this study compared to the reference years. However, since it is very unlikely that all patients analyzed within this study developed oral malignant lesions right at the beginning of the lockdown and did not attend a medical examination exactly over the period of 3 months, this effect will probably not have become apparent in view of the relatively small group size in 2020 (213 patients).

Despite the observed level of compensation in the healthcare of patients with oral malignant lesions in Germany within the analyzed periods, it is necessary to establish plans for the future to ensure appropriate care of patients at risk. While maintaining hygiene measures, individuals identified as high-risk patients (triage) should be encouraged to adhere to a regular recall in order to detect malignant transformation of precursor lesions at an early stage and initiate interventions. Close monitoring of patients with other risk constellations (alcohol and nicotine consumption) should also be considered. In particular, detailed information about the signs and symptoms of oral cavity carcinoma as well as the necessary hygienic measures and the weighing of the corresponding risks are of great importance. Here, telemedicine could also offer a substantial support for the education and empowerment of the patient. In an extensive study that analyzed more than 2.5 million social media posts, Moraliyage et al. found that the COVID-19 pandemic significantly affected the emotional well-being of patients with cancer. The fears were mainly related to the topics of therapy, mental health, diagnosis and continuation of therapy and concerned the delay of diagnosis of diseases, an interruption of diagnostic procedures already started, the postponement of cancer screening appointments, the lack of access to healthcare as well as the delay of follow-up examinations [42]. A study on head and neck cancer patients from Germany revealed that 40% were anxious regarding their disease but also in regard to the psychological and physical consequences of the pandemic. Social isolation (prohibition of visits) was reported to be a main problem (58.5%) [43,44,45].

The current pandemic and future similar scenarios demand the necessary restructuring of the healthcare system in order to be able to take preventive measures and adequately treat patients with malignant diseases even under difficult conditions and to keep the number of infections as low as possible. This is also of great relevance from a health economic point of view in light of the massive therapy costs, which will continue to rise in the future due to individualized treatment regimens. The emotional impairment of patients with cancer and the associated concerns about their care, which became clear in the study by Moraliyage et al., show not only the structural requirements of risk-adapted healthcare but also the relevance of emotional support for patients. The importance of digitalization and especially telemedicine can be seen here. With the help of this, it may be possible to inform patients at an early stage about signs and symptoms of malignant conditions and the relevance of preventive examinations, the risk of transmitting infections and to strengthen the patient in his or her self-care; if necessary, also to provide support from other disciplines. Furthermore, oral cancer screenings performed by dentists play a crucial role in the early detection of premalignant and malignant lesions in the oral cavity [46] and, therefore, it should be notified for future epi- or pandemic settings that withholding dental care has an impact on oral disease progression.

Limitations of this study are the bias connected with the retrospective character of the data collection (e.g., selection bias) as well a possible seasonal influence on hospital admissions of patients with oral malignant lesions due to a fluctuation of surgeons or references from external dentists which may vary over different periods. Another limitation is the relatively small sample size of 653 patients in total and the regional limitation to three medical centers in Germany that may not be representative for hospitals in other regions and the German population as a whole. From a statistical point of view, the analytical pathway with only few time points is statistically limitedly robust. By large, this study is descriptive in nature and should be interpreted with caution.

## 5. Conclusions

Within these limitations and pooled over all three study centers, we found that the incidence of oral malignant lesions did not increase in association with the COVID-19 2020 lockdown or the post lockdown period when compared to previous year. There was a trend towards a shorter time to intervention during the lockdown period which in turn supports the theory that the German healthcare system was still able to manage important interventions adequately despite implemented restrictions. Notably, in the federal state with the strictest lockdown implementation, Rhineland-Palatinate, there was an unusually high number of low tumor stages in the post lockdown period compared to previous years, possibly as primary practitioners had their practices closed, leading to a delayed detection and referral, highlighting the relevance of primary dental care for early diagnosis of premalignant and malignant lesions.

## Figures and Tables

**Figure 1 cancers-13-02892-f001:**
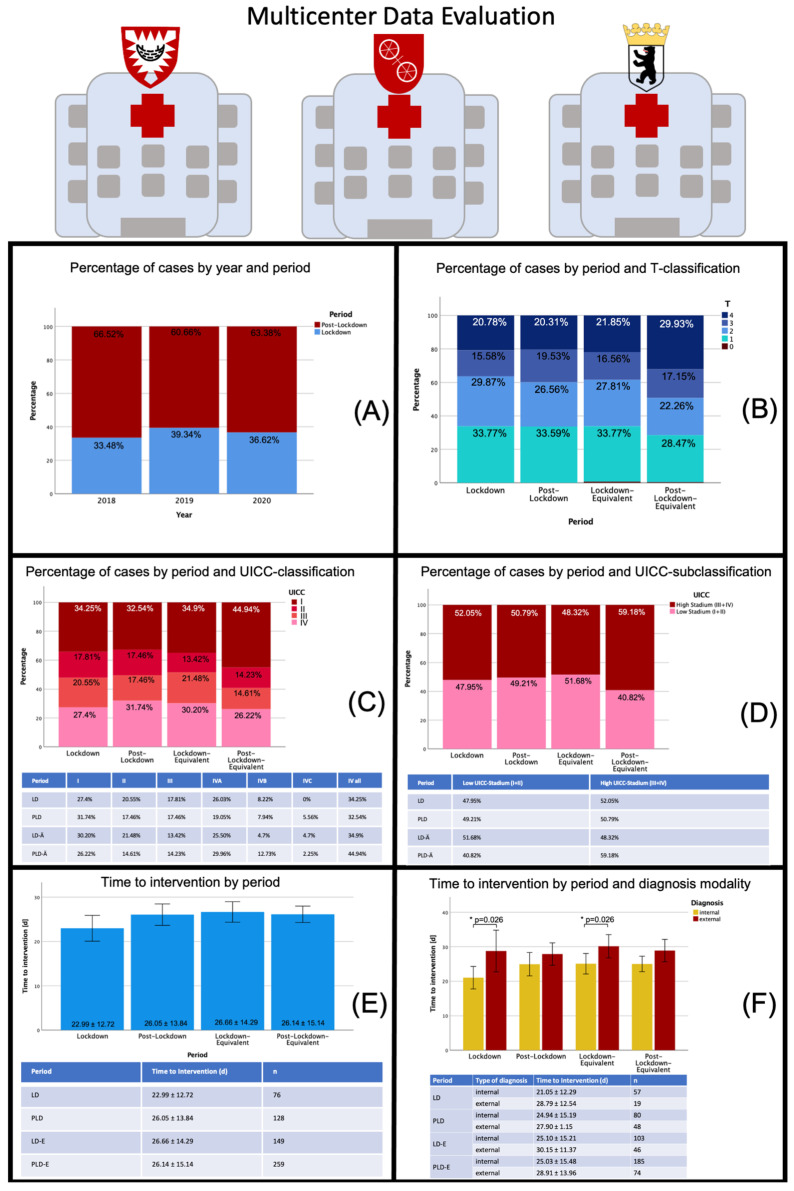
Data sheet multicenter data evaluation: Distribution of patients with oral malignant lesions by different periods, distribution of tumor stages (T-classification, UICC-classification) and time to intervention during the different periods. Abbreviations: LD—lockdown; PLD—post-lockdown; LD-E—lockdown equivalent; PLD-E—post-lockdown equivalent; UICC—Union international contre le cancer. In (**A**) cancer diagnoses per year and period are displayed (in percent) showing a stable distribution in both regards. In (**B**) the distribution of cancer cases by T-classifications divided into the different periods are displayed. Nearly no difference could be detected between LD and PLD, whereas a higher proportion of T4 cancer could be seen during the PLD period in 2020 compared to the previous years. This effect is also reflected in the UICC-classification (**C**) and UICC-subclassification (**D**) with more stage IV cancer during the PLD period in 2020 (50.79% versus 59.18%). In (**E**) the time to intervention for the different time periods is displayed showing a trend towards a shortened time to intervention during the 2020 LD period compared to previous years while the time in the 2020 PLD period remained stable compared to previous years. In (**F**) time to intervention was further divided by diagnosis modality. A shorter time to intervention by internal diagnosis during the LD period in 2020 was observed. * *p* < 0.05.

**Figure 2 cancers-13-02892-f002:**
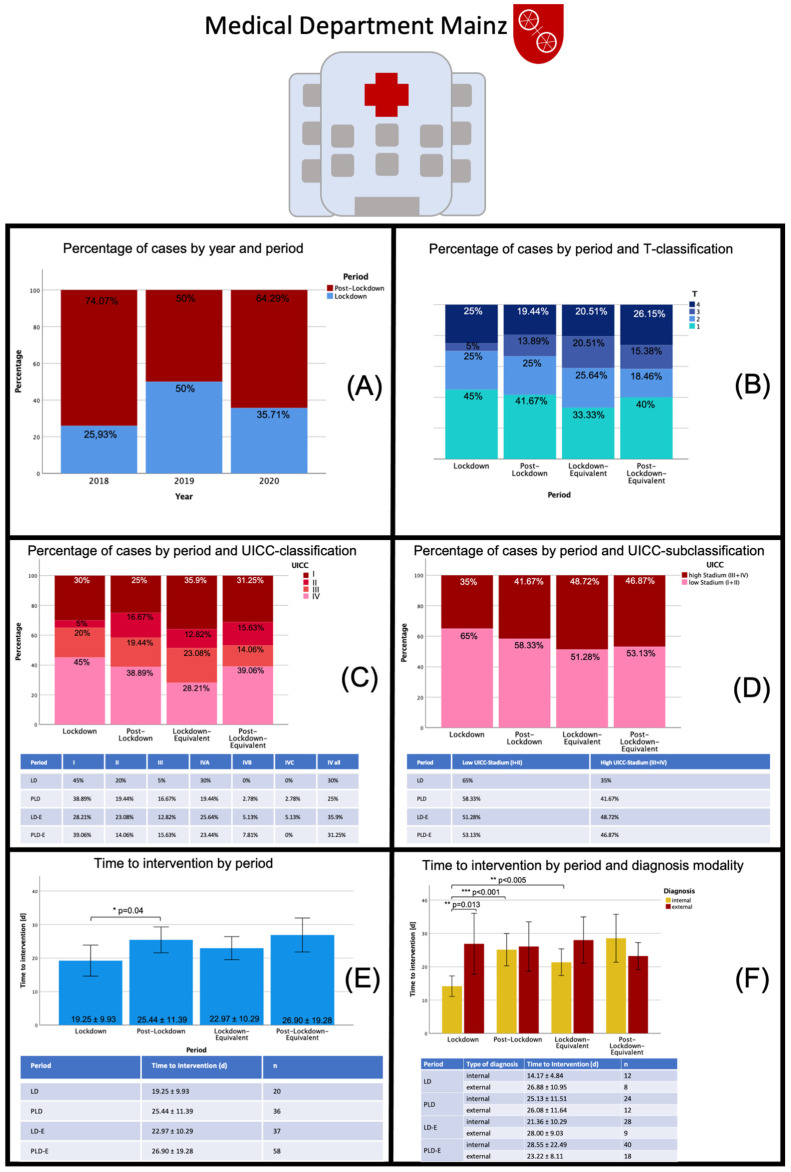
Data sheet Medical Department Mainz: Distribution of patients with oral malignant lesions by different periods, distribution of tumor stages (T-classification, UICC-classification) and time to intervention during the different periods. Abbreviations: LD—lockdown; PLD—post-lockdown; LD-E—lockdown equivalent; PLD-E—post-lockdown equivalent; UICC—Union international contre le cancer. In (**A**) cancer diagnoses per year and period are displayed (in percent) showing a varying distribution in both regards. As such, there is no obvious trend in 2020 compared to previous years in terms of patient number distribution. In (**B**) the distribution of cancer cases by T-classifications divided into the different periods are displayed. In the LD period 2020 both, more T1 and T4 cancer stages were diagnosed compared to the LD-E period, whereas the distribution between the PLD period and the PL-E period did not differ much. This effect is also reflected in the UICC-classification (**C**) and UICC-subclassification (**D**). Here, a trend towards a higher percentage of smaller cancer stages was evident when comparing the LD period in 2020 to the reference years. The subgroup analysis also found more patients with lower tumor stages (UICC I and II) within the LD period in 2020 compared to the previous years. In (**E**) the time to intervention for the different time periods is displayed showing a statistically significant difference between the LD and PLD period in 2020 (*p* = 0.04). In (**F**) time to intervention was further divided by diagnosis modality. A statistically significant shorter time interval for internal diagnosis during LD 2020 was seen (*p* = 0.013). Contrary, time to intervention for internal diagnosis increased again from LD to PLD (*p* < 0.001). * *p* < 0.05; ** *p* < 0.005; *** *p* < 0.001.

**Figure 3 cancers-13-02892-f003:**
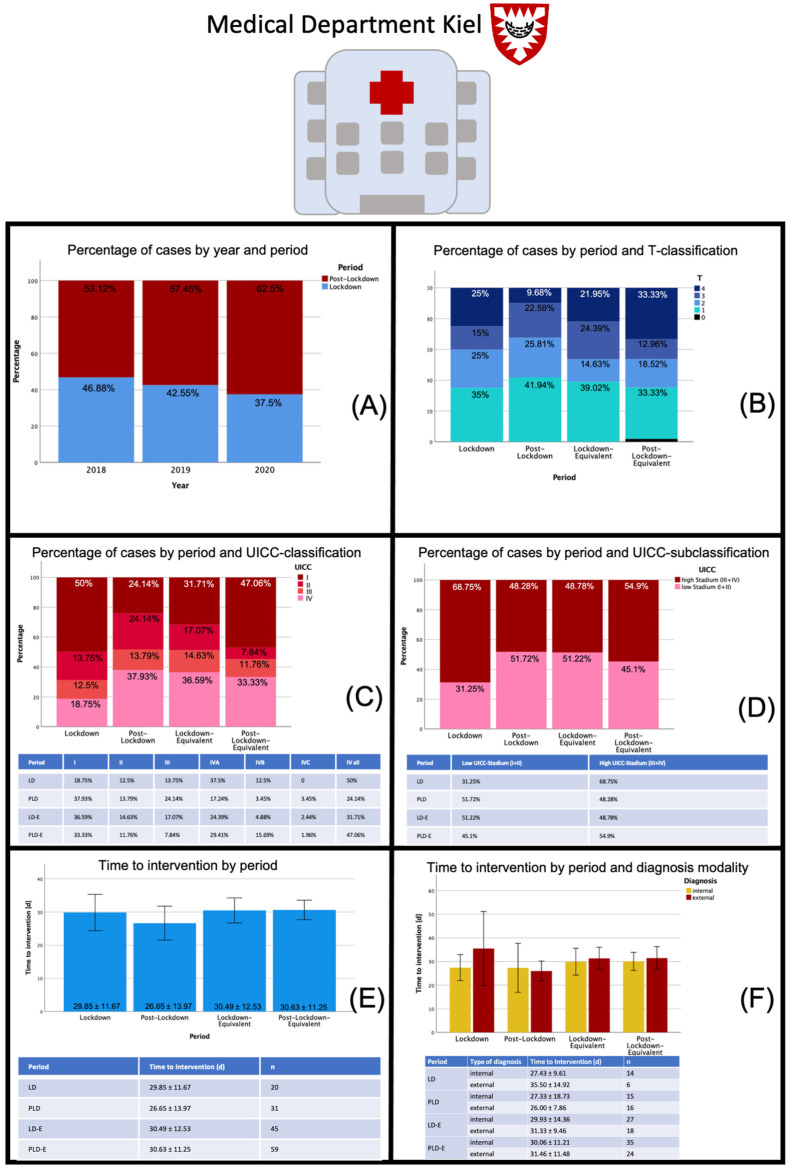
Data sheet Medical Department Kiel: Distribution of patients with oral malignant lesions by different periods, distribution of tumor stages (T-classification, UICC-classification) and time to intervention during the different periods. Abbreviations: LD—lockdown; PLD—post-lockdown; LD-E—lockdown equivalent; PLD-E—post-lockdown equivalent; UICC—Union international contre le cancer. In (**A**) cancer diagnoses per year and period are displayed (in percent) showing a slight trend towards fewer cases during the LD period over years which, however, is not relevant for the comparison of 2020 with previous years. In (**B**) the distribution of cancer cases by T-classifications divided into the different periods are displayed. In the PLD-E period more T4 carcinomas were diagnosed, whereas the number of intermediate sized (T2, T3) carcinomas showed reduced values compared to the LD period in 2020. The UICC-classification (**C**) and UICC-subclassification (**D**) yielded interesting results. In 2020 a significantly higher proportion of stage IV carcinomas (50% versus 31.71%) in the LD period was observed, while the proportion of stage I carcinomas decreased significantly. Considering the relatively stable results of the T-classification, this must indicate a higher rate of lymph nodes and distant metastases in the population. In contrast, the number of high tumor stages in the 2020 PLD period was similar to previous years (51.72% versus 45.1%). In (**E**) the time to intervention for the different time periods is displayed showing no significant change in time to intervention in the 2020 LD period compared with previous years, while it was slightly, but not significantly, lower in the 2020 PLD period compared with 2018 and 2019. In (**F**) time to intervention was further divided by diagnosis modality. No statistically significant difference could be shown in time to intervention between internal and external diagnosis.

**Figure 4 cancers-13-02892-f004:**
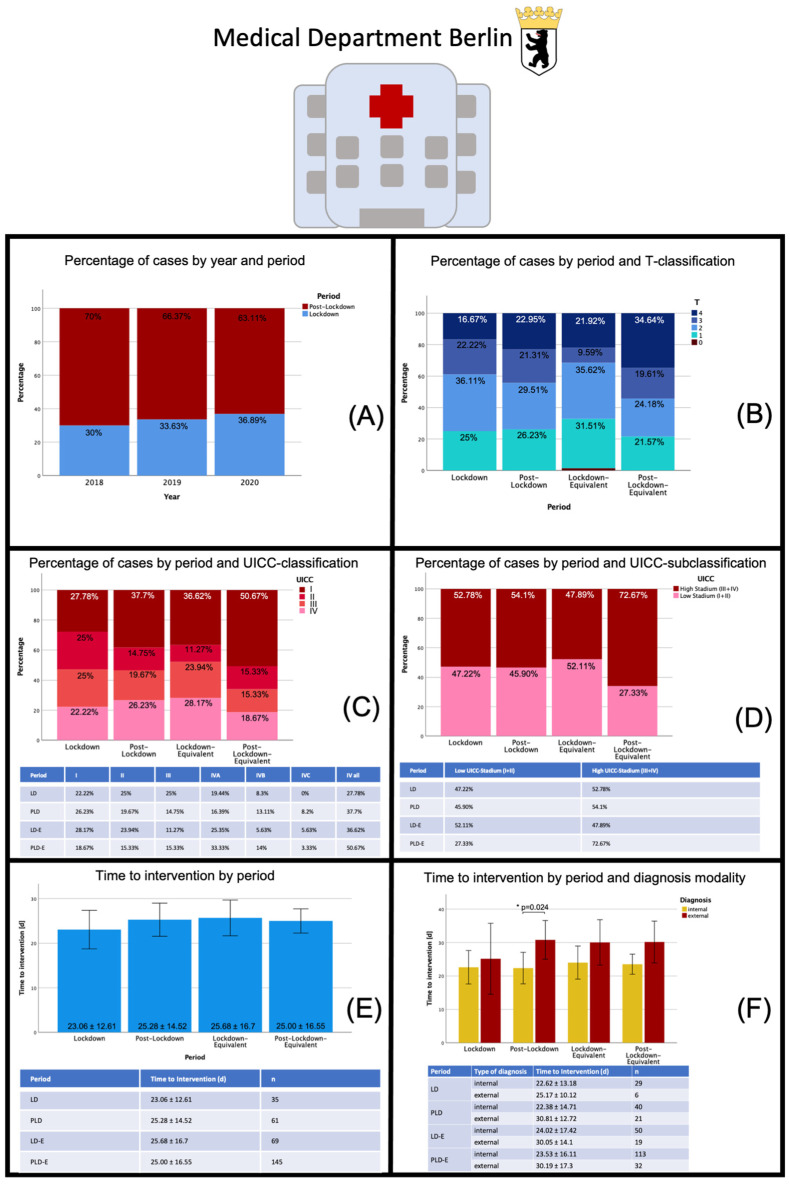
Data sheet Medical Department Berlin: Distribution of patients with oral malignant lesions by different periods, distribution of tumor stages (T-classification, UICC-classification) and time to intervention during the different periods. Abbreviations: LD—lockdown; PLD—post-lockdown; LD-E—lockdown equivalent; PLD-E—post-lockdown equivalent; UICC—Union international contre le cancer. In (**A**) cancer diagnoses per year and period are displayed (in percent) showing a stable distribution in both regards. In (**B**) the distribution of cancer cases by T-classifications divided into the different periods are displayed. In the LD period 2020 fewer T1 and more T3 carcinomas were diagnosed, whereas the number of T4 carcinomas decreased during the PLD period in 2020 compared to the previous years. This effect is also reflected in the UICC-classification (**C**) and UICC-subclassification (**D**). In (**E**) the time to intervention for the different time periods is displayed showing a slightly lower time to intervention during LD 2020. In (**F**), time to intervention was further divided by diagnosis modality. Within the PLD period in 2020, the time to intervention for internal diagnosis was significantly lower than that of the external diagnosis. * *p* < 0.05.

## Data Availability

Data is contained within the article and Appendix A. The raw data is available from the corresponding author on reasonable request.

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
