# Peer review of "Consequences of the COVID-19 Pandemic and Governmental Containment Policies on the Detection and Therapy of Oral Malignant Lesions—A Retrospective, Multicenter Cohort Study from Germany"

_cancers, 2021, doi:10.3390/cancers13122892_

Round 1

Reviewer 1 Report

In this submission, Heimes et al. investigate a potential screening delay and its consequences in patients with oral cancer during the pandemic. This is an interesting study. This manuscript can be accepted with subject to major revisions. Following are specific comments;

  • Authors should give a very brief introduction to SARS-CoV-2 and the proposed use of oncology drugs for its treatment. This will give more insights into its relevance to oncology. Furthermore, it will give basic information to readers. Following paper can be cited in these regards; https://doi.org/10.1016/j.nantod.2020.100962
  • Line 93: 'Even so, 93 at present, there is no specific biomarker to detect precursor lesions and early stages of ...' needs a bit more explanation and reference(s).
  • section 2.1, the study is about cases in Germany while in paragraphs 'line 99' authors discuss case study from the UK. I suggest adding some case studies from Germany to make it relevant to their work.
  •  Figure 1-4: the captions must be extended.

Author Response

We would like to thank the reviewer for the constructive comments on our work.

Comment #1
Authors should give a very brief introduction to SARS-CoV-2 and the proposed use of oncology drugs for its treatment. This will give more insights into its relevance to oncology. Furthermore, it will give basic information to readers. Following paper can be cited in these regards; https://doi.org/10.1016/j.nantod.2020.100962
Answer: We added information regarding SARS-CoV-2 and the use of nanotechnology for its treatment. See lines 52–59 and 161–164

Comment #2
Line 93: 'Even so, at present, there is no specific biomarker to detect precursor lesions and early stages of ...' needs a bit more explanation and reference(s).
Answer: We added more information on this topic as can be seen in lines: 101–122

Comment #3
section 2.1, the study is about cases in Germany while in paragraphs 'line 99' authors discuss case study from the UK. I suggest adding some case studies from Germany to make it relevant to their work.
Answer: Thank you for this constructive comment, we added some information on case studies in Germany as can be seen in lines 126–128, 134–149 and 153–157

Comment #4
Figure 1-4: the captions must be extended.
Answer: We extended the captions as suggested.

Reviewer 2 Report

 This manuscript deals with the impact of lockdown on oral cancer surgery. When it comes to cancer treatment, it is a high-priority disease, if not urgent. Therefore, I think this study is valuable information as the infection situation of COVID-19 is still uncertain all over the world.

 As you described, during these lockdown period, the usual oral surgery such as tooth extraction and cyst removal might be postponed. When the lockdown is over, such diseases must be operated. If so, how did you deal with such disease? Or will minor surgery be postponed than usual?

 It also states that the dental office plays an important role for the detection of malignancies, but is there any possibilities that the patient himself noticed it because he was concerned about his oral health during the lockdown?

Author Response

We would like to thank the reviewer for the constructive comments on our work.

Comment #1
As you described, during these lockdown period, the usual oral surgery such as tooth extraction and cyst removal might be postponed. When the lockdown is over, such diseases must be operated. If so, how did you deal with such disease? Or will minor surgery be postponed than usual?
Answer: As described by the University Cancer Center Hamburg, “In terms of interdisciplinary care for solid cancer patients, the surgical departments have dramatically reduced their elective operation programs while at the same time ensuring immediately necessary cancer surgery to be maintained.”. With decreasing incidence and in accordance with the requirements of the universities, they gradually returned to regular service, so that elective surgery could also be performed. Of course, in addition to tumor surgery, surgical therapy was also performed for other urgent indications, such as infectious diseases (abscesses) and fractures. The patients were then triaged according to the risk of infection and the urgency of the surgery. We added a section explaining this fact in lines 180–184.

Comment #2
It also states that the dental office plays an important role for the detection of malignancies, but is there any possibilities that the patient himself noticed it because he was concerned about his oral health during the lockdown?
Answer: We added a section explaining the relevance of dentists in early detection of malignancies in lines 116–122. Even though we cannot exclude the presence of isolated cases, the significantly reduced number of physician contacts as well as the significantly reduced number of initial cancer diagnoses by pathologists indicate that a visit to the physician's office/clinic occurred rather less frequently. However, increased awareness of potentially malignant lesions due to limited access to health care is conceivable and would be plausible given the reported cases of fear of tumor progression. However, the data rather point to an opposite trend.

Round 2

Reviewer 1 Report

I am pleased to recommend the revised manuscript for publication in Cancers.